# Impact of Physical Activity on DNA Methylation Signatures in Breast Cancer Patients: A Systematic Review with Bioinformatic Analysis

**DOI:** 10.3390/cancers16173067

**Published:** 2024-09-03

**Authors:** Chantalle Moulton, Veronica Lisi, Monica Silvestri, Roberta Ceci, Elisa Grazioli, Paolo Sgrò, Daniela Caporossi, Ivan Dimauro

**Affiliations:** 1Unit of Biology and Genetics of Movement, Department of Movement, Human and Health Sciences, University of Rome Foro Italico, 00135 Rome, Italy; chantalle.moulton@sanraffaele.it (C.M.); veronica92lisi@gmail.com (V.L.); m.silvestri5@studenti.uniroma4.it (M.S.); daniela.caporossi@uniroma4.it (D.C.); 2Unit of Biochemistry and Molecular Biology, Department of Movement, Human and Health Sciences, University of Rome Foro Italico, 00135 Rome, Italy; roberta.ceci@uniroma4.it; 3Unit of Physical Exercise and Sport Sciences, Department of Movement, Human and Health Sciences, University of Rome Foro Italico, 00135 Rome, Italy; elisa.grazioli@uniroma4.it; 4Unit of Endocrinology, Department of Movement, Human and Health Sciences, University of Rome Foro Italico, 00135 Rome, Italy; paolo.sgro@uniroma4.it

**Keywords:** DNA methylation, breast cancer, exercise, epigenetic

## Abstract

**Simple Summary:**

Breast cancer significantly affects women globally, but physical activity (PA) has been shown to improve the quality of life, aid recovery, and enhance survival rates in patients. Studies reveal that PA influences DNA methylation, both globally and at gene-specific levels, potentially reversing abnormal methylation linked to cancer. This review compiles research on PA’s impact on DNA methylation in breast cancer patients. The findings suggest that PA increases global DNA methylation in tumour tissues and alters gene-specific promoter methylation across various genes. Bioinformatic analysis indicates that these genes are primarily involved in metabolic pathways, cell cycle regulation, mitosis, cellular stress responses, and diverse binding processes. The Human Protein Atlas supports these findings, showing gene functionality in 266 tissues, including various breast tissues. Overall, PA’s ability to modify DNA methylation patterns in breast cancer patients may aid in the restoration of normal cellular functions and enhance recovery and survival outcomes.

**Abstract:**

Breast cancer (BC) continues to significantly impact women worldwide. Numerous studies show that physical activity (PA) significantly enhances the quality of life, aids recovery, and improves survival rates in BC patients. PA’s influence extends to altering DNA methylation patterns on both a global and gene-specific scale, potentially reverting abnormal DNA methylation, associated with carcinogenesis and various pathologies. This review consolidates the findings of the current literature, highlighting PA’s impact on DNA methylation in BC patients. Our systematic analysis indicates that PA may elevate global DNA methylation within tumour tissues. Furthermore, it appears to modify gene-specific promoter methylation across a wide spectrum of genes in various tissues. Through bioinformatic analysis, to investigate the functional enrichment of these affected genes, we identified a predominant enrichment in metabolic pathways, cell cycle regulation, cell cycle checkpoints, mitosis, cellular stress responses, and molecular functions governing diverse binding processes. The Human Protein Atlas corroborates this enrichment, indicating gene functionality across 266 tissues, notably within various breast tissues. This systematic review unveils PA’s capacity to systematically alter DNA methylation patterns across multiple tissues, particularly in BC patients. Emphasising its influence on crucial biological processes and functions, this alteration holds potential for restoring normal cellular functionality and the cell cycle. This reversal of cancer-associated patterns could potentially enhance recovery and improve survival outcomes.

## 1. Introduction

Breast cancer (BC) stands as the most commonly diagnosed cancer among women, leading to a significant number of cancer-related deaths [1,2]. BC accounts for 23% of diagnosed cancer cases and 14% of cancer-related deaths, posing a considerable health challenge [2]. Currently, the incidence rate of BC is on the rise, attributed to advancements in diagnostic approaches and screenings that enhance accurate detection. Notably, the reported mortality rate has not escalated proportionately, suggesting an improved capacity in clinical management to cope with BC [3].

Numerous lifestyle factors have been identified as contributors to BC risk, such as smoking, alcohol consumption, poor-quality diet, and low levels of physical activity (PA) [4,5,6,7]. Adequate levels of PA have demonstrated the potential to reduce BC risk, improve cancer-specific mortality, and positively impact overall well-being [2,6,7,8,9,10,11,12,13,14]. The protective effects of PA are attributed to multiple mechanisms, including immune function, inflammation, growth factors, sex hormones, and epigenetic modifications [8,11,13,15,16,17,18,19].

In epigenetics, DNA methylation plays a crucial role in BC carcinogenesis, influencing both global and gene-specific levels [16,17,18,20]. DNA methylation, an epigenetic process involving the addition of a methyl group to the 5′ position of the cytosine pyrimidine ring within cytosine-phosphate-guanine (CpG) dinucleotides, represents a flexible genomic parameter capable of influencing gene expression and genome function.

The literature reflects a growing interest in global DNA methylation and gene-specific DNA methylation at CpG sites in genes associated with BC [21,22]. Global DNA hypomethylation, a decrease in DNA methylation over large genomic regions that are typically methylated, is associated with carcinogenesis [23,24]. The implications of hypomethylation depend on the genomic regions involved, with hypomethylation within gene bodies associated with aberrant gene expression [25,26,27]. In BC, gene-specific DNA methylation in the promoter regions of tumour suppressor genes (TSGs) plays a key role in carcinogenesis through transcriptional silencing [20,28,29]. Studies have shown that TSGs such as RASSF1A, APC, RARβ2, HIN1, and H-cadherin are more commonly methylated in BC than in adjacent non-malignant breast tissue [30,31,32,33,34,35]. 

PA significantly impacts DNA methylation patterns, affecting both global and gene-specific methylation across various tissues [36,37,38]. At the global level, PA can lead to changes in overall DNA methylation levels, crucial for maintaining genomic stability and regulating gene expression [37,38]. These changes may help in silencing repetitive elements and modulating the activity of genes involved in essential cellular processes [39,40]. On a gene-specific level, PA can alter the methylation status of particular genes, influencing their expression and potentially affecting many pathways, including those related to metabolism, inflammation, and stress response [37,38,41]. The mechanisms through which PA exerts these effects are not fully understood but likely involve modifications in the activity of DNA methyltransferases (DNMTs) and ten-eleven translocation (TET) enzymes [38]. Overall, the relationship between PA and DNA methylation highlights the potential of exercise to influence epigenetic marks and promote overall health and well-being.

While the evidence supporting the relationship between DNA methylation and BC, both in carcinogenesis and prognosis, is becoming increasingly clear, the exploration of how PA can modify these epigenetic modifications in BC is still in its early stages [42,43,44,45,46].

Therefore, our aim is to systematically review the existing literature regarding PA and DNA methylation in BC patients. This was carried out by investigating modifications of DNA methylation status on both a global and gene-specific level following both intervention and observational studies in BC populations. Moreover, we further examined the effects of PA-induced DNA methylation on different biological aspects in BC patients by studying functional enrichment using bioinformatic analysis. Our results highlight the importance of PA in re-establishing and preserving some of the biological functions in BC patients undergoing medical treatment.

## 2. Methods and Materials

### 2.1. Study Selection

This systematic review was conducted using PRISMA guideline and registered on PROSPERO (CRD42022276532). The literature search was conducted on PubMed, Scopus, and Web of Science (last accessed on 4 April 2024). The performed search strategy was generated using the following terms, ((Breast Cancer) AND (DNA Methylation OR DNMT OR Epigenetics OR methyltransferase OR promoter methylation OR methylome) AND (Exercise or Physical activity)), and adapted to suit each respective database as their syntax rules required (Table 1).

All articles that contained the searched keywords in the title, abstract and/or keyword sections were examined against the selection criteria. Additionally, references from relevant articles were manually searched to identify further eligible studies.

After the search results were generated, studies were selected if they were deemed eligible, based on predetermined criteria, independently by two different reviewers. All articles not relevant to the topic and any articles that did not contain original research were discarded. Duplicate articles were also excluded.

The selected articles focused on the effects of exercise or physical activity (PA) on DNA methylation in relation to breast cancer. These effects were investigated through in vivo human models conducted in populations of BC patients. In order to exhaustively investigate the topic, a variety of measures for both DNA methylation and PA parameters were included in this systematic review, as there is a paucity of studies investigating this topic in BC populations, and moreover, a lack of overlap between methodologies amongst those available. PA parameters included exercise interventions, measures of fitness, and self-reported physical activity levels—either in daily life or retrospectively in personal history.

PA parameters were restricted to measurements that could be evaluated individually without confounding variables, such as integration into a quality-of-life score. DNA methylation parameters were measured in levels of either global or gene-specific methylation status, including methylome measurements.

Exclusion criteria were applied to articles not relevant to the topic, studies in populations not diagnosed with breast cancer, articles not originally written in English, or studies where PA levels/exercise were not clearly investigated as independent modulators of DNA methylation.

The titled, abstract, and keywords of all eligible articles were further screened, and a final selection was made by the reviewers. The reviewers compared and discussed their independent first selections to create an agreed-upon final set of results after reviewing the full texts. There were no disagreements between reviewers regarding the study eligibility.

### 2.2. Data Collection Process 

All data included in this systematic review were obtained from the original articles (i.e., tables, Supplementary Tables, Supplementary Data/Files).

The findings of the selected articles were analysed and categorised based on the nature of each PA parameter in the selected articles—whether these were planned intervention or reported PA levels. All data points of interest were categorised to analyse and compare the relevant methods and results.

The data items of interest included the population analysed, the methods used for methylation analysis, the sample used for analysis, and the resulting modifications in the DNA methylation status of either specific genes or global DNA. These modifications were examined as a consequence of PA levels and/or cardiovascular fitness levels.

In all studies analysing gene-specific promoter methylation, specific primers were designed in CpG-rich regions within known gene promoters [10,44,46,47].

### 2.3. Functional Profiling and Tissue Enrichment Analysis

All genes with a modulated, gene-specific DNA methylation status resulting from PA, measured in any tissue, were compiled from the collective studies and utilised for functional enrichment analysis. In order to conduct a bioinformatic analysis, measures of global DNA methylation were not included; only measures of gene-specific DNA methylation that were significantly altered by PA were included. The bioinformatic analysis methodology outlined by Reimand and colleagues [48] was modified to meet our specific requirements. Specifically, g:Profiler (https://biit.cs.ut.ee/gprofiler/ (accessed on 30 April 2024)) was used for functional pathway enrichment, using g:GOSt functional profiling, including Biological Processes (BPs), Molecular Function (MF), Reactome Pathways (RPs), and Kyoto Encyclopedia of Genes and Genomes (KEGG). Tissue enrichment was also conducted using the Human Protein Atlas (HPA) tools (https://www.proteinatlas.org/ (accessed on 5 May 2024). The resulting adjusted *p*-values (*p* < 0.05) were utilised to construct histograms of -log10(adj. *p*-value) to visually represent the most significant pathway and tissue enrichments.

## 3. Results

### 3.1. Search Results

As represented in Figure 1, the search strategy generated a total of 233 articles: 30 papers were found on PubMed, 73 on Scopus, and 130 on Web of science by using the respective search strategies for each database. After the removal of duplicates, 171 papers remained for abstract screening. During the initial selection, based on title, abstract, and keywords review, 26 articles were deemed eligible to be reviewed at full-text level. Of these 26 articles, 6 were considered appropriate for inclusion after full-text review, while 20 articles were excluded as they did not meet the complete inclusion criteria.

### 3.2. Studies Characteristics

This systematic review is composed of six articles addressing how DNA methylation is modified by PA in in vivo human models. All study characteristics and findings of interest are categorised below based on the structure of each study (exercise intervention vs. self-reported PA levels) (Table 2). In total, data from about 3042 female subjects were included. The age range of patients was from ~20 to 98 years. However, it should be taken into account that the age of most of the patients recruited in each study analysed was between 45 and 60 years. At the beginning of the experimental protocol, participants had low levels of physical activity (mean duration physical activity < 150 min/week), particularly, in longitudinal studies. The basal level of physical activity was 21.8 ± 38.0 min/week in one study [10], <150 min/week in one study [46], and <90 min/week in one study [43], whereas in the cross-sectional studies women were classified as inactive (<9.23 h of RPA/week) and active (≥9.23 h of RPA/week) or inactive (<6.36 h of RPA/week) and active (≥6.36 h of RPA/week) [44,47,49]. Therefore, all subjects recruited in the selected studies were defined as sedentary according to the guidelines suggested by World Health Organization (WHO, https://www.who.int/europe/publications/i/item/9789240014886 (accessed on 1 June 2024) [50]).

The exercise modalities and PA forms, as well as their respective measurement modalities, varied between studies. These included exercise interventions, self-reported PA levels and exercise frequency, and VO_2max_ assessment. Three studies performed exercise interventions, and three studies carried out measurements of self-reported PA levels, while one study also measured cardiovascular fitness through a VO_2max_ assessment. The exercise intervention in Gorski et al. [43] was structured as a 5-month program of treadmill-based endurance training, practiced three times per week, which was aimed at increasing VO_2peak_, while Zeng et al. [10] used a 6-month program of moderate-intensity exercise at 150 min/week (primarily brisk walking on a graded treadmill). In addition, Gorski et al. measured cardiorespiratory fitness (VO_2peak_) using a treadmill-based, symptom-limited cardiopulmonary exercise test, using a stepwise modified Balke protocol until exhaustion [43]. In Moulton et al. (2024), the PA intervention spanned over 16 weeks, consisting of combined strength and aerobic training sessions twice per week on non-consecutive days, totalling 32 sessions [46]. 

The self-reported studies measured PA levels by mostly using questionnaires. McCoullough et al. [44,47,49] used interviews and a modified version of a physical activity frequency questionnaire (PAFQ) from Bernstein et al. [51] to measure recreational physical activity (RPA) 2–3 months after original BC diagnosis. 

The tissue origins of the samples analysed across all studies varied between tumour samples, white blood cells, and skeletal muscle. Gorski et al. extracted DNA from skeletal muscle tissue biopsies [43], Moulton et al. from blood [46], and Zeng et al. from both blood and tumours [10]. McCullough et al., in 2015 [44] and in 2017 [47], extracted DNA from archived formalin-fixed, paraffin-embedded tumour tissue of first primary BC, and Mccullough and colleagues, in 2015 [49] and in 2017 [47], extracted DNA from white blood cells. McCullough et al., in 2015 [44] and in 2017 [47], analysed gene-specific methylation in tumour tissue using MSP to analyse 3 genes (ESR1, PR, and BRCA1), and the MethyLight assay to measure a further 10 genes. The authors also analysed global DNA methylation in white blood cell DNA [47,49] using the luminometric methylation assay (LUMA), a quantitative measurement of genome-wide DNA methylation, as described by Bjornsson et al. [52], and LINE-1, where four CpG sites in the promoter region of LINE-1 were assessed using a validated pyrosequencing-based methylation assay [53]. Gorski et al. analysed the methylome by using Infinium MethylationEPIC BeadChip Array, followed by analysing differentially methylated positions (DMPs), conducting pathway enrichment analysis (KEGG pathways), and analysing differentially methylated regions (DMR) [43]. Zeng et al. used a microarray (Infinium HumanMethylation27 BeadChip) that analyses 27 578 CpG sites in 14 495 genes, to analyse DNA methylation in blood and qMSP to analyse L3MBTL1 promoter methylation in tumour samples [10]. Finally, Moulton et al. utilised qMSP to analyse the promoter methylation of the genes SOD1, SOD2, Catalase, L3MBTL1, RASSF1A, and BRCA1 [46]. 

To date, there is no universally defined classification of methylation status. Indeed, each study published and present in this systematic review utilised a different strategy to demonstrate that their results were robust and reliable. In particular, Zeng et al. [10] and Moulton et al. [46] set a *p*-value < 5 × 10^−2^ (two-sided) and <5 × 10^−5^ (two-sided), respectively, which were considered statistically significant for the MSP and microarray data on methylation, to conservatively select genes for further analysis. As suggested from other authors [54], McCullough and colleagues [44,47,49] utilised a 4% cut-off as the percentage of the methylated reference and/or a coefficient of variation < 1%. Gorski and colleagues [43] set a *p*-value < 0.01, and an average median methylated and unmethylated signal above 11.5, as recommended in the Oshlack workflow [55].

### 3.3. Study Populations and Treatment Descriptions

Gorski et al. (2023) conducted a randomised controlled exercise training trial that included BC patients randomised into either an exercise group or control group, with a third arm of age-matched women with no prior history of cancer who also performed the same exercise training program. The BC survivors had been diagnosed with stage II-III HER2-negative BC, at the age of/below 60 years, and included those who had been treated with anthracycline-based chemotherapy [43]. 

Zeng et al. (2012) recruited BC survivors who were diagnosed with stage 0-IIIA BC, had completed adjuvant therapy a minimum of 6 months prior to the study, and were randomised into an exercise group and a control group [10]. While this intervention study did not specify the stage or subtype of the tumour, these results were compared with those of frozen tumour samples from a different cohort of patients, considering tumour stage and ER status, categorised by L3MBTL1 expression levels [10]. 

Moulton et al. (2024) recruited stage I-III BC patients, without differentiating based on tumour subtype, who were randomised into a control group (45.15 ± 5.54 years of age) and an exercise group (50.55 ± 5.69 years of age) after undergoing surgery but before starting adjuvant therapy (including chemotherapy, hormone therapy and radiotherapy) [46]. 

Of note is that the studies by McCullough et al. (2015a, 2015b and 2017) were conducted using resources from the Long Island Breast Cancer Study Project, with either case–control or follow-up components [44,47,49]. This project included BC patients who were newly diagnosed with first primary in situ or invasive breast cancer between 1 August 1996 and 31 July 1997 and who were between the ages of 20–98 at the time of diagnosis, and no specifics regarding medical treatments were stated in the articles [44,47,49].

### 3.4. Results of Studies with Exercise Interventions

Three studies conducted exercise interventions, which are often prospective studies, which are specifically tailored to evaluate direct impacts of an assigned exposure on specific outcome measures [56]. Zeng et al. identified the significant modification of the DNA methylation status of 43 genes that were altered in BC patients after a 6-month moderate-intensity aerobic exercise intervention [10]. Specifically, when comparing the methylation patterns before and after the 6-month exercise intervention, of these 43 genes with significant changes in methylation levels (*p* < 0.05), the most significant genes were EPS15 (*p* = 1.27 × 10^−7^, 3% increase in methylation after exercise), RP11-450P7.3 (*p* = 4.65 × 10^−7^, 4% decrease in methylation), and KIAA0980 (*p* = 4.676 × 10^−6^, 2% decrease in methylation). The largest increase in methylation was in CXCL10 (*p* = 2.876 × 10^−5^, 5% increase), and the largest decrease was in ABCB1 (*p* = 1.64 × 10^−5^, 8% decrease) [10]. Of these 43 genes, L3MBTL1, a tumour suppressor gene, showed a decrease in DNA methylation status after exercise, accompanied by an increase in L3MBTL1 gene expression, which possibly contributes to an increased rate of survival in BC patients [10]. This was accompanied alongside the altered methylation of other genes with possible links to BC, such as MSX1 [10]. 

In blood, Moulton et al. identified PA-induced changes in the promoter methylation of SOD1, SOD2 and L3MBTL1 after 16 weeks of training [46]. Specifically, PA was effective in reducing the promoter methylation of SOD2 (Exercise group PRE vs. POST, 18.915 ± 3.947 vs. 15.188 ± 3.424% 5mC, *p* = 0.002), and L3MBTL1 (Exercise group PRE vs. POST, 53.613 ± 8.057 vs. 39.946 ± 6.987% 5mC, *p* = 0.0005) in the BC training group. Furthermore, in the control group, BC patients without exercise training, levels of SOD1 promoter methylation increased at the end of experimental protocol (Control group PRE vs. POST, 0.042 ± 0.012 vs. 0.079 ± 0.022% 5mC, *p* < 0.0001) and when compared with the same experimental point in the exercise group (POST: CG vs. EG, 0.079 ± 0.022 vs. 0.034 ± 0.011% 5mC, *p* < 0.0001), whereas physical activity was able to prevent this increase as SOD1 promoter methylation levels were maintained in the exercise training group. These methylation changes were inversely linked with corresponding gene expression [46]. Furthermore, the authors also measured physical and fatigue-related parameters, which were used in regression analysis to identify relationships between methylation changes and physical function improvements. They identified that increases in SOD1 and catalase promoter methylation were linked to increased physical fatigue, and that decreases in SOD2 distal promoter methylation were linked with improvements in a 6 min walk test, reflecting improved physical fitness [46].

In skeletal muscle, Gorski et al. found that a 5-month exercise intervention in BC patients reversed the hypermethylated patterns seen in cancer survivors to patterns resembling healthy age-matched controls in 300 promoter-associated CpG islands [43]. Specifically, training in BC survivors had a stronger effect on the genome than in healthy-age matched controls, as 14,215 CpG sites were differentially methylated after training in BC survivors compared with 2149 DMPs after training in controls. Training exerted a predominantly hypermethylation trend in both BC survivors (8586 hyper- vs. 5629 hypo-methylated DMPs) and in the healthy age-matched controls (1745 hyper- vs. 404 hypo-methylated DMPs), in which hypermethylation predominantly occurred in regulatory regions. Moreover, regarding CpG islands in promoter regions, these were largely changed to hypomethylated signatures in response to exercise training, where 99.9% of DMPs (2358 hypo- vs. 2 hyper-methylated) in CpG islands became hypomethylated in BC survivors due to training [43]. Furthermore, the authors identified 972 DMRs in BC survivors after training, with 298 of these DMRs located in CpG islands in promoter regions [43]. Their analysis of DMRs shows that certain genes (BAG1, BTG2, CHP1, KIFC1, MKL2, MTR, PEX11B, POLD2, S100A6, SNORD104, and SPG7) were hypermethylated in BC survivors, while training was able to revert these genes back to a hypomethylated signature [43]. These exercise-induced DNA methylation modifications were also stable for up to 10 years later. In addition, pathway enrichment analysis of their results showed that exercise training in cancer survivors was able to induce a hypomethylated signature in pathways associated with DNA replication/repair, the cell cycle, transcription, translation, proteosome and mTOR signalling [43]. 

### 3.5. Results of Studies with Self-Reported PA Levels 

Three studies conducted retrospective studies by using measures that were self-reported by the volunteers; this protocol involved using an instrument comprising questions on the attributes of the subjective parameters being measured [57]. McCullough et al. identified that postmenopausal BC patients with moderate and high levels of RPA had increased global DNA methylation in breast cancer measured using LUMA analysis, where an increase in the LUMA–breast cancer association was observed among postmenopausal BC survivors who were recreationally physically active (moderate RPA (≤9.23 h/wk): OR = 2.62; 95% CI = 1.44, 4.75 and high RPA (>9.23 h/wk): OR = 2.62; 95% CI = 1.53, 4.49). [49]. Furthermore, McCullough et al. also showed, in 2015, that women with higher levels of RPA (>9.23 h/wk) were more likely to have ER + PR + breast tumours with methylated GSTP1 (OR = 2.33, 95% CI 0.79–6.84) [44]. McCullough et al. later identified, in 2017, an association between pre-diagnostic RPA and all-cause mortality, whereby all-cause mortality was decreased only in recreationally physically active women with methylated promoters of CCND2 (HR 0.56, 95% CI 0.32–0.99), APC (HR 0.60, 95% CI 0.40–0.80), HIN (HR 0.55, 95% CI 0.38–0.80), and TWIST1 (HR 0.28, 95% CI 0.14–0.56) in tumours, but not among those with unmethylated tumours, with a significant interaction (*p* < 0.05) [47]. 

### 3.6. Effects of Physical Activity on Recovery and Survival Outcomes

The study by Zeng et al. (2012) compared changes in blood DNA methylation and gene expression due to PA with those in frozen tumour samples from a cohort of patients, who were followed up after an average of 86.3 months. As seen in the volunteers who performed the PA intervention, the authors compared the increased L3MBTL1 gene expression levels with those in tumours and identified that patients that had tumours with high L3MBTL1 levels also had improved survival at patient follow-up [10]. 

Moulton et al. (2024) underscored the benefits of PA in further improving clinical symptoms like pain, fatigue, body composition, and QoL, while also reporting in a preceding study that PA led to improvements in markers of inflammation and markers of oxidative stress [19,46]. Although they have no follow-up data, these studies show a beneficial effect of PA on patient QoL and clinical markers of recovery, thereby reinforcing its therapeutic value in cancer recovery [19,46].

In the study by Gorski et al. (2023), both trained and untrained BC survivors had a higher baseline DNA methylation age compared to healthy controls, with no significant difference between the BC groups. After the training period, trained cancer survivors showed a trend towards reduced DNA methylation age, but this was not statistically significant. Furthermore, Gorski et al. (2023) indicated that PA was associated with reduced mortality and improved survival among BC patients. However, they did not differentiate these effects between trained and untrained cancer patients. The overall trend suggested that PA had beneficial effects on patient survival rates and mortality outcomes [43].

In their 2017 study, McCullough et al. further analysed medical records at a 5-year follow-up and mortality at a follow-up after approximately 15 years. Here, they noted that women who performed RPA across their lifespan tended to have a lower BMI and were less likely to have nodal involvement, while they found little difference in other clinical characteristics (i.e., ER or PR status) among physically active women compared with inactive women. Furthermore, the authors found that women who had higher levels of RPA had lower all-cause and BC-specific mortality [47].

### 3.7. Bioinformatics Pathway Analysis

To understand the possible downstream effects of the observed DNA methylation changes in the promoter regions of genes due to PA, we observed the functional pathway and tissue enrichments due to our resulting modulated DNA methylation signatures through the results of our selected articles. The results included in the bioinformatics analysis included all articles with significant gene-specific DNA methylation changes, excluding measures of global DNA methylation, thereby leaving five out of the six selected articles to be included in the analysis [10,43,44,46,47]. The results, generated using g:profiler, reflect the systemic effects of PA, as all analysed tissues were included in the analysis, which focused on MFs, BPs, KEGGs, Reactome pathways, and HPA tissue enrichment. Figure 2 represents histograms of the -log_10_(adj. *p*-value) of the top/top 25 significantly modulated pathways, as well as the tissues in which it was mostly enriched, while a list of all significantly enriched pathways and tissues can be found in Appendix A. 

The most significantly enriched MFs primarily highlighted various binding functions, including “Protein binding” (GO:0005515), “mRNA binding” (GO:0003729). and “Enzyme binding” (GO:0019899), as well as “Catalytic activity” (GO:0003824) and “translation initiation factor activity” (GO:0003743). The resulting enriched BPs were generally focused on “metabolic processes” (GO:0008152), such as “Organic substance metabolic process” (GO:0071704), and “Nitrogen compound metabolic process” (GO:0006807), as well as on the cell cycle and cell division, e.g., “Cell cycle” (GO:0007049), “Mitotic cell cycle process” (GO:1903047) and “Cell division” (GO:0051301), and also included “Cellular response to stress” (GO:0033554). Enriched Reactome pathways predominantly highlighted the cell cycle and cell cycle checkpoints, such as “Cell Cycle” (HSA-1640170), “Cell Cycle, Mitotic” (HSA-69278), and “Cell Cycle Checkpoints” (HSA-69620). Moreover, KEGG pathways were significantly enriched in “Cell cycle” (hsa04110) and “DNA replication” (hsa03030), with some being implicated in diseases such as “Parkinson disease” (hsa05012) and “Hepatocellular carcinoma” (hsa05225). In addition, the HPA option within g:profiler was used to analyse possible the tissue enrichment in the PA-modulated genes we identified, in order to extrapolate tissues that may have been affected by the changes in methylation. This showed that our genes may have been active in 266 different tissues. Of note is that in terms of female reproductive tissue, several were enriched in various tissues in the breast (HPA:0050000, HPA:0050051, HPA:0050091, HPA:0050052), cervix (HPA:0630000), ovary (HPA:0340000), fallopian tube (HPA:0210000), and endometrium (HPA:0160051, HPA:0170000, HPA:0160000), while some were also enriched in the colon (HPA:0130000), stomach (HPA:0540000), oesophagus (HPA:0190221), and most other tissues. 

## 4. Discussion

This systematic review aimed to consolidate the existing evidence on the impact of PA on DNA methylation in BC patients. The findings collected and presented in this review indicate that PA can influence both global and gene-specific DNA methylation, potentially contributing to improved clinical outcomes for BC patients.

### 4.1. Global DNA Methylation

Two of the six articles [47,49] analysed markers of global DNA methylation, using LUMA, a methylation sensitive restriction assay, or LINE-1, which uses PCR-based methods to estimate the methylation of major genomic repeat elements [58,59,60]. McCullough et al. found increased global DNA methylation levels (using LUMA) in BC patients with higher self-reported postmenopausal RPA levels [49]. However, in 2017, McCullough et al. found no significant associations in self-reported RPA levels with either LUMA or LINE-1 analysis [47]. 

In the mammalian genome, most of the DNA methylation is in repetitive elements, such as transposons [61,62]. Transposable elements amount to roughly 45% of the human genome and include long and short interspersed nuclear elements (LINE and SINE, respectively), long terminal repeats (LTR), retrotransposons, and DNA transposons. These sequences have the potential to interfere with gene expression regulation and genome structure through deletions, insertions, inversions, and translocations of genomic sequences. However, this potential for damage is reduced when repetitive sequences are silenced via CpG methylation [61,62,63]. 

Multiple studies have shown associations between global hypomethylation and BC carcinogenesis, [23,24], and that chronic PA maintains levels of global DNA methylation [16,17,18]. While one study did not confirm the positive association between PA and global DNA methylation [47], none found contrastingly negative associations. McCullough et al. [49] support a protective effect of PA against BC through the augmentation and maintenance of global DNA methylation levels, preventing the global hypomethylation seen in BC carcinogenesis [23,24,49,64]. Increased global DNA methylation levels may reduce the amount of aberrant gene expression in tumour-related genes, otherwise expressed due to global hypomethylation [23,24,25,26,27].

### 4.2. Gene-Specific DNA Methylation

Five out of the six selected articles measured DNA methylation on a gene-specific level [10,43,44,47]. Aberrant gene-specific DNA methylation is strongly linked with BC, and the effects are largely dependent on the genome context and the function of the genes. The number of genes analysed varied significantly, reflecting differences in study design and research focus. Studies examining a larger number of genes may provide a broader overview of DNA methylation changes but may lack the depth of analysis seen in studies focusing on a smaller, more targeted set of genes. Conversely, studies with a narrow focus often provide detailed insights into specific gene pathways or mechanisms but may miss broader methylation patterns. 

McCullough et al. [44] showed that higher self-reported postmenopausal RPA levels were associated with ER + PR + tumours when GSTP1 was methylated, possibly due to its role in detoxification reactions, contrasting to the systematic response to PA that increases the production of reactive oxygen species (ROS)—potentially showing an increased risk of BC due to postmenopausal RPA in ER + PR + breast tumours [44,65]. McCullough et al. later published a study in 2017 showing that in physically active women, there were significant decreases in the promoter methylation of the BC-related genes APC, CCND2, HIN1, and TWIST1, showing a beneficial effect of PA on DNA methylation patterns [47]. 

Of the 43 genes that Zeng et al. found with significantly modulated methylation statuses after a six-month exercise intervention in BC patients, the expression of 6 genes were significantly correlated with overall survival [10]. Of these ‘survival’-correlated genes, the expression of three genes (GLUD1, L3MBTL1 and MSX1) was consistent with the identified trend in PA-induced methylation modifications, showing a potential mechanism whereby PA reduces gene-specific methylation, resulting in increased gene expression and better survival [10]. The methylation status of L3MBTL1, a TSG, was significantly negatively correlated with the gene expression of L3MBTL1, supporting a possible PA-induced mechanism protecting against BC [10,66,67]. Consistent with the work in Zeng et al., Moulton et al. also identified significant decreases in L3MBTL1 promoter methylation in response to PA, supporting the presence of a mechanism whereby PA may protect against BC by decreasing L3MBTL1 promoter methylation, allowing for the expression of the TSG [10,46]. Moulton et al. also identified that PA was able to decrease SOD2 promoter methylation and prevent an increase in SOD1 promoter methylation, which was seen in the control group. BC medical treatment usually induces a high level of stress, especially in the form of oxidative stress, not only in the cancerous tissue, but systematically as well [19,68,69,70,71]. The effect of PA on SOD1 and SOD2 promoter methylation may assist the body in combatting the systemic oxidative stress side effects of BC treatment, by enabling the expression of SODs [46].

### 4.3. Long-Term Effects of PA on DNA Methylation

Gorski et al. showed that even 10–24 years after BC diagnosis and treatment, cancer survivors present with increased promoter CpG hypermethylation in skeletal muscle compared to healthy age-matched controls, and that PA was able to reset the methylation signature to that of healthy controls by demethylating 2358 genes across the genome [43]. This was especially the case within genes that form pathways related to mitosis, cell cycle, transcription, and the proteosome [43]. This is consistent with the studies in the literature that show that aerobic, high-intensity, and resistance exercise have demethylating effects in skeletal muscle [72,73,74,75,76]. Moreover, both chronic and acute exercise have been shown to result in hypomethylation in human and mouse skeletal muscle [72,73,74,75,76].

### 4.4. Functional Impact of PA-Induced DNA Methylation Changes

Bioinformatic analysis of genes with PA-modulated gene-specific DNA signatures revealed that these changes may predominantly impact pathways important in the cell cycle, cell cycle regulation, transcription, cell division, and metabolism (Figure 2), which are all pathways and functions necessary for proper cell function and division, promoting healthy cells and tissues and preventing cancer. The list of significant functional enrichments, along with all implicated genes, can be found in the Appendix A. Many significantly enriched Reactome pathways focused on cell cycle checkpoints. These allow for the important restoration of normal cell cycle regulation, as these checkpoints are largely inactivated in cancers, causing uncontrolled cell growth (reviewed in [77]). Moreover, the most significantly enriched BPs focus on metabolic processes. In BC, there is a form of metabolic reprograming that is observed, aiding in the progression and proliferation of cancerous cells during carcinogenesis [78,79]. The increased functioning of metabolic process BPs due to PA may restore normal metabolic processes, promoting healthy cells through the demethylation of vital genes. 

To date, the extent to which PA can affect DNA methylation patterns across different tissues and the consistency of these patterns across tissues remain unknown [80]. The comparison of DNA methylation signatures across tissues, with some plagued by cancer and some not, e.g., blood and skeletal muscle, has become of interest in finding potential BC risk biomarkers as a less invasive alternative to breast tissue biopsies. Therefore, analysis of BC-related DNA methylation in liquid biopsies, i.e., blood samples, has garnered interest. Zeng et al. found consistent methylation patterns of L3MBTL1 in both breast tumour and blood DNA, showing that those in the blood and breast mirror each other at BC diagnosis [10]. With a lack of follow-up analysis of the breast tissue after PA, it is unknown if PA affects the breast in the same way as it affects the blood. However, as reported by Gorski et al., PA modulated DNA methylation across the genome in many promoter regions within skeletal muscle, showing that PA modulates DNA methylation in tissues besides the blood [43]. Comparisons of DNA methylation signatures across tissues have shown strong correlations in DNA between blood and saliva, saliva and buccal tissues, blood and brain, saliva and brain, and blood and breast [10,81], although there is a paucity of research that compares PA-induced DNA methylation changes across tissues.

However, PA exerts beneficial effects within BPs, MFs, Reactome pathways and KEGGs, which may ensure proper cell functioning and prevent cancer. While our selected studies measured DNA methylation in different tissues, the HPA shows significant enrichment across many tissues, exceeding those that were analysed in the articles. Relevant to BC are the various breast tissues and other female reproductive tissues. While these tissues have not been analysed for their effects of PA on DNA methylation, one cannot rule out that similar changes occur in these tissues as well, following the trend that has been observed in studied tissues [80]. If these changes are systematically present, they could result in improved recovery and prevention on the BC level, and improved systemic health, improving quality of life.

### 4.5. Mechanisms Underlying PA-Induced DNA Methylation Changes

The exact mechanism underlying changes in DNA methylation in response to PA remains not yet fully understood [82]. PA induces a plethora of stimuli in vivo, which could lead to changes in DNA methylation (Figure 3). 

PA has been shown to affect the levels and activities of enzymes related to the regulation and maintenance of DNA methylation, including the TET family and DNMTs. Bryan et al. speculated that the physiological consequences of PA modify DNA methylation by inhibiting DNMT function (including DNMT1, DNMT3a, and DNMT3b), preventing the methylation that leads to carcinogenesis and gene instability [42]. McGee et al. speculated that the PA-induced changes in DNA methylation are due to the regulation of DNMTs and enzymes involved in DNA demethylation [82]. In various cell lines, TET1 was induced by hypoxia [82,83]. The effects of PA on redox homeostasis and ROS levels may also explain the effects on DNA methylation changes, as ROSs are implicated in modulating DNA methylation by targeting the expression/activity of DNMTs and TETs, through reducing the availability of the cofactor S-Adenosyl methionine (SAM) and preventing the reduction of Fe(III) back to F(II) [84,85]. 

Moreover, Jeltsch and colleagues identified the optimal flanking sequences that affect DNMT and TET binding/activity in order to methylate/demethylate cytosines [86,87,88], suggesting that genomic context impacts changes in methylation regulating enzymes and may explain how exercise can have differing hyper/hypomethylating effects. 

It is also not yet understood how PA has effects across different tissues, in terms of whether the stimulus that alters methylation occurs within each affected tissue, or if it occurs in specific PA-affected tissues and effectors are then transported to other tissues via cross-tissue communication to later enforce changes, such as through the transport of DNMT transcripts [89], NRF2, a transcription factor that has been shown to increase the transcription of DNMTs [90], or microRNAs that target DNMTs [91,92], which may affect recipient cells after transport, for example via extracellular vesicles [93]. However, these mechanisms have not been investigated in the context of systematic alterations in DNA methylation induced by PA in BC patients. 

## 5. Study Limitations

This systematic review has several limitations. Given the emerging nature of the research topic, only a small number of studies met our selection criteria. Many of these studies are observational or involve self-reporting, which can introduce participant bias. The PA interventions and measurements vary widely, leading to minimal overlap between studies. Each study tends to focus on different aspects (PA modality, DNA methylation measurement, or tissues of interest), resulting in a lack of consistent findings.

The DNA methylation analyses also show considerable methodological diversity, with few comprehensive methylome-wide analyses to evaluate gene-specific modifications thoroughly. This methodological variance makes direct comparisons between studies difficult. Consequently, our synthesis provides an overview of PA effects on DNA methylation in BC patients, but its results should be interpreted with caution. The heterogeneity in study designs suggests that our conclusions are part of a broader, varied body of evidence rather than definitive findings.

Additionally, most studies do not distinguish between BC subtypes, despite certain subtypes having specific methylation signatures. The variability in the number of genes studied can lead to statistical errors: studies examining many genes may face a higher risk of type I errors (false positives), while those focusing on fewer genes may face a risk of type II errors (false negatives), potentially missing significant changes.

Gene selection is often based on prior evidence and research interests, which can introduce bias and limit the comprehensiveness of the findings. This selective focus reduces overlap and comparability across studies, complicating the drawing of consistent conclusions. Notably, only one gene (L3MBTL1) was assessed in more than one study included in this review.

Furthermore, although the reviewed studies adopt moderate-intensity exercise protocols, it is not clear whether or not the methylation data are normalised to the possible variations in blood cell composition and in serum volume the end of experimental protocol. This could confound certain gene-specific results in the case of lineage-specific methylation, where different cell types have distinct methylation patterns. Additionally, not all studies provided detailed information on the DMPs or their genomic locations that were investigated, which would have allowed for comparison studies, and gene expression data were inconsistently available. 

Another limitation is the lack of detailed information on the correlation between genetic and molecular changes due to PA and clinical outcomes such as tumour size reduction, symptom alleviation, and life expectancy. Many studies also lack detailed data on patients’ treatment and disease statuses, making it challenging to attribute observed genetic changes solely to PA interventions.

Finally, global DNA methylation analyses have inherent biases, as the methylation status of certain regions may have specific functional implications that cannot be evaluated in a global analysis. This issue is compounded by the diverse methodologies used across studies, further complicating the functional interpretation of the results [25].

## 6. Future Implications for Research

Future studies should prioritise chronic exercise interventions with quantitative measurements over observational studies using self-reported PA levels to reduce participant bias. Research should compare different PA modalities, such as cardiovascular versus resistance training, to determine which is more effective in modulating DNA methylation status. Comparing DNA methylation across different tissue types, especially within breast tissue, is also essential to identify common modulation patterns.

Studies should conduct methylome-wide analyses to understand the lasting effects of PA on DNA methylation and whether these changes revert to BC signatures over time. Assessing DNA methylation both before and after BC onset within the same individuals would provide deeper insights into the timeline and association of these changes with breast carcinogenesis.

To improve study comparability and build a cohesive body of evidence, researchers should agree on a defined list of specific genes to be routinely assessed in PA, DNA methylation, and BC research. This consensus would facilitate more reliable comparisons across studies and enable meta-analyses to be conducted. Combining broad genome-wide analyses with targeted approaches can allow researchers leverage the strengths of both methods. Initial broad screenings can identify candidate genes and pathways, which can then be validated and studied in detail using targeted approaches. Future blood-based studies should correct for blood cell composition in methylation analyses, if relevant, to ensure changes are directly attributable to physical activity. Additionally, researchers should prioritise detailed DMP annotations and include gene expression analyses to better understand the functional consequences of methylation changes, allowing for study comparability.

Establishing a core set of genes for analysis, based on current evidence and research consensus, would enhance the comparability of future studies and provide a clearer understanding of the effects of PA on DNA methylation in BC patients. This approach would address current challenges in gene-specific methylation studies and guide future research to enhance the robustness and comparability of findings in this field.

## 7. Conclusions

While the mechanisms remain unclear, it is reasonable to speculate that PA has substantial, wide-reaching, and systemic effects on DNA methylation signatures in BC patients, across various tissues (e.g., blood, skeletal muscle and breast tumour), and at both a gene-specific and global level (See graphical abstract). Here, we show that genes with a PA-induced modulation of DNA methylation in BC patients, measured across multiple tissues, were largely focused on healthy cell functioning and the regulation of cellular processes, which allowed for normal cell functioning and a normal cell cycle—as opposed to the trends seen in cancer. For instance, modifying biological processes related to metabolism and cell cycle regulation in breast cancer patients could significantly improve treatment outcomes. These changes could reduce tumour growth, increase sensitivity to therapies, improve programmed cell death, and prevent drug resistance. Better-regulated metabolism can also weaken tumour cells and enhance immune response. Overall, these improvements could lead to slower disease progression, longer survival, and better quality of life, while also paving the way for more personalised treatment strategies. Therefore, PA may contribute positively to BC recovery through the alteration of DNA methylation, rectifying DNA methylation signatures that previously may have contributed to BC carcinogenesis, and beneficially affecting DNA methylation signatures in various tissues to potentially meet the stress induced during BC recovery.

## Figures and Tables

**Figure 1 cancers-16-03067-f001:**
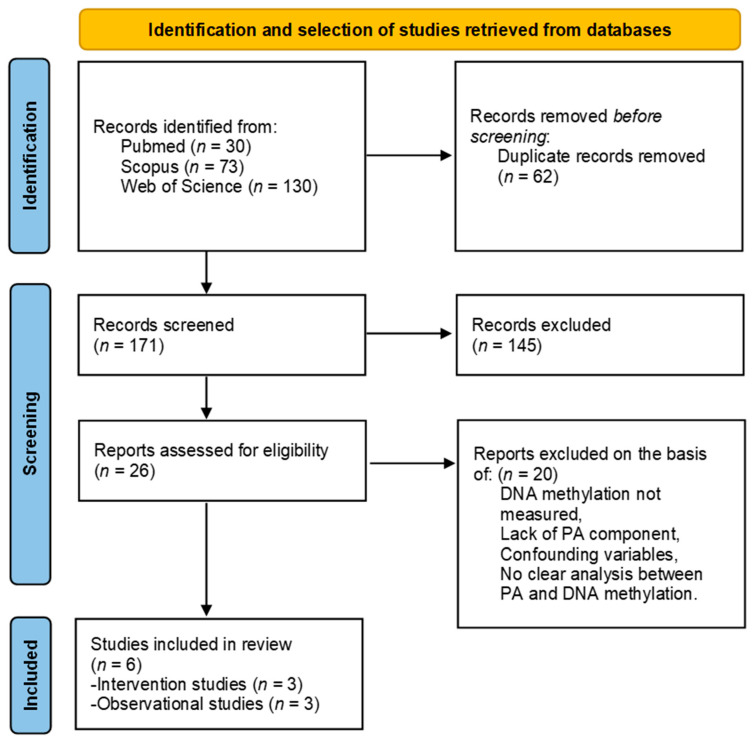
Flowchart demonstrating the article selection process.

**Figure 2 cancers-16-03067-f002:**
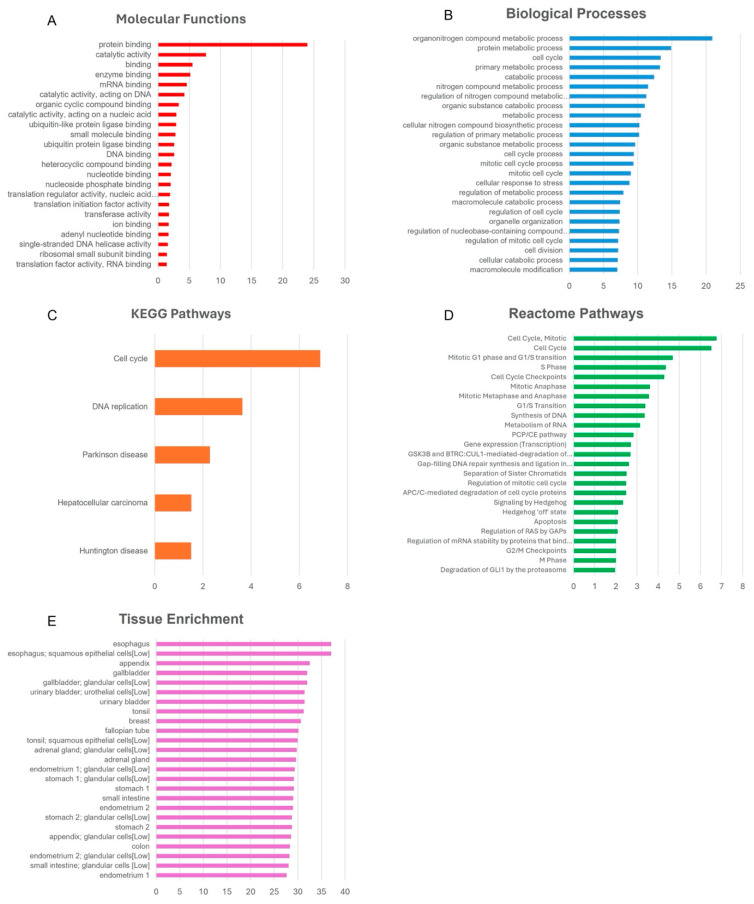
Histograms showing the results of the functional pathway enrichment analysis, conducted using g:profiler, of the genes with a significantly modulated DNA methylation status due to PA in BC populations. (**A**) Top significantly enriched molecular functions, (**B**) top 25 significantly enriched biological processes, (**C**) top 25 significantly enriched Reactome pathways, (**D**) top significantly enriched KEEG pathways, (**E**) and top 25 significantly enriched tissues reflected by the Human Protein Atlas.

**Figure 3 cancers-16-03067-f003:**
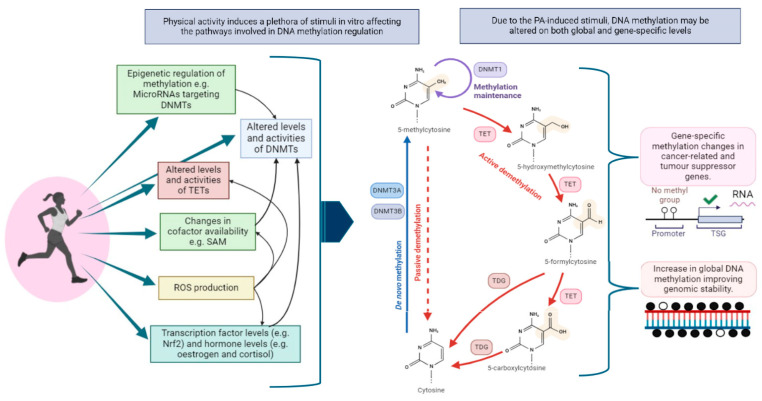
Hypothetical mechanisms underlying PA-induced DNA methylation. SAM, S-Adenosyl methionine; DNMTs, DNA methyltransferases; TET, ten-eleven translocation enzymes; ROS, reactive oxygen species; Nrf2, nuclear factor erythroid 2-related factor 2; TDG, thymine-DNA glycosylase.

**Table 1 cancers-16-03067-t001:** Research queries used on each respective database.

Database	Query
Pubmed	((Breast cancer [Title/Abstract]) AND (exercise [Title/Abstract] OR Physical activity [Title/Abstract]) AND (DNA Methylation [Title/Abstract] OR DNMT [Title/Abstract] OR Epigenetics [Title/Abstract] OR methyltransferase [Title/Abstract] OR promoter methylation [Title/Abstract] OR methylome [Title/Abstract]))
Scopus	(TITLE-ABS-KEY (“breast cancer”)) AND (TITLE-ABS-KEY (exercise OR “physical activity”)) AND (TITLE-ABS-KEY (“dna methylation” OR epigenetics OR methyltransferase OR “promoter methylation” OR methylome OR dnmt)) AND (EXCLUDE (DOCTYPE, “re”))
Web of Science	((ALL = Breast cancer) AND (ALL = exercise OR ALL = Physical activity) AND (ALL = DNA Methylation OR ALL = Epigenetics OR ALL = methyltransferase OR ALL = promoter methylation OR ALL = methylome OR ALL = DNMT))

**Table 2 cancers-16-03067-t002:** Effects of PA on DNA methylation patterns in BC populations. Table summarising the study design and results of interest from all selected articles that evaluated the effects of PA on DNA methylation signatures in BC patients. DNA methylation results are presented using the gene codes of the genes associated and/or affected by the PA-induced methylation changes reported by the original authors.

Intervention Studies						
Author	Study Design	Population Sample Size	Sample	Exercise Intervention	Methylation Detection Technique and Analysed Target	Significant Modulation(s) in DNA Methylation as a Result of PA/Exercise
Gorski et al., 2022 [43]	Two-armed randomised controlled exercise training trial	Breast cancer survivors diagnosed with stage II-III HER2-negative breast cancer Cancer trained (*n* = 7, 62.5 ± 4.2 age of years) Cancer-untrained (*n* = 6, 59.9 ± 5.7 age of years) Healthy age-matched, trained (*n* = 10, 57.6 ± 4.0 age of years)	Skeletal muscle	5 months of three-times -per-week treadmill-based endurance training, aimed at increasing VO2peak	Infinium MethylationEPIC BeadChip array: Methylome DMPs and DMRs	DMPs: RAD51L3, ZFR, SIRT2, PPWD1, TMEM146, EIF4G3, UQCRC1, INO80C, KIAA0226, ROBLD3, CCDC28A, IFT80, JOSD1, AP1G2, PPP4R3B, CHMP5, COX7A2L, ZNF641, NXT1, GTF3C1, HIST1H4A, DNAJA3, COQ10B, C16orf46, CARHSP1, C11orf48, BUB1B, ALDH4A1, MYADM, DFFA, C14orf178, RNF145, CBR4, IMMP2L, INTS10, OAZ2, TRAIP, RBM26-AS1, R3HDM1, LZTR1, CYP2U1, WDR60, MTCH2, WDR19, GANC, SIDT2, C4orf14, PTRH2, SLC35A3, HACD3, RBL2, AP2A1, SHMT1, TGFB1, SNORA6, NDUFB11, YPEL1, ZNF394, CELSR2, CDCA7L, MYH10, TSPAN33, KIFC1, G3BP1, ING1, VKORC1, CHTF8, MDH1, DIDO1, ATF7, SLC25A37, SLC38A10, SIVA1, ATPIF1, NPHP4, LDHA, NAGK, MCM3, UBE3B, CLP1, RRM2B, ATXN1L, BANP, PRSS27, CDKN2D, TYMS, ZNF570, DHPS, GNE, HIST1H4B, SMCR7L, RTN4, CBARA1, ZKSCAN5, OTUD7B, KCTD21, CLNS1A, ATP5L, ZDHHC5, GSK3B, ELOVL6, DIS3, ACIN1, NUMB, FAM151B, PNPLA6, SGOL1, PRMT7, ZNF594, MTR, TOMM34, TMEM170A, SOD1, NDUFV2, MORN1, RHBDD2, MKL1, ZNF282, LOC100329108, ESD, DMAP1, TMPO, SNORD43, SURF6, ZFYVE1, LOC400027, NFYA, PELI3, PIP4K2C, TEX2, BUB1B, LMBRD1, NDUFS2, ST5, ARL8B, UBA52, UBL5, ZNF337, MYO9A, AFG3L1, ALG14, DMXL1, CCNC, KIFC1, MYBBP1A, ZFP62, C6orf147, DLG4, C21orf91, C9orf40, C11orf80, SARM1, PFDN4, SPATS2, ZNF222, TYMP, PEX5, NDEL1, PTGES3, TNFAIP3, PSMC3IP, PAF1, FANCG, PGM1, SCYL3, PIGY, ACTG1, TRA2A, PIGP, TCTN2, C3orf71, NAA15, SPG7, USE1, DPM3, IMPA2, ACAD10, HNRNPA0, C1orf112, RNASEN, SMC3, CAST, TAF15, FRMD8, TAF5, RPL36AL, COL4A3BP, C1orf35, GMPR, MTBP, ANKRD31, BBS2, PHF12, NDUFA9, CLNS1A, HACE1, CD320, OTUD3, AURKB, P4HTM, C15orf29, CORIN, EXOSC4, CKAP2, MAP3K8, URM1, STAT5B, SEL1L, KLC4, AMZ2, TOR2A, PODNL1, AGFG1, LOC284900, ESCO1, DNAJC28, SSR2, CENPV, MAK16, EXD2, USP21, ELMO2, PYROXD1, TRIM24, BUB1, REXO2, ZNF143, EIF4G1, GNRHR2, MCM3APAS, CASC1, SRBD1, PLBD2, TPCN1, CSRP1, HS3ST3B1, ANKRD9, GIGYF2, COL4A3BP, PGRMC2, ICMT, C19orf43, NIPA2, RNFT1, CCDC102B, DCPS, BCL2L2-PABPN1, TACO1, FAM46A, RB1CC1, KBTBD4, ARF4, BTRC, EWSR1, HAUS8, NDUFA2, CAPNS1, CASP6, SETD5, TSNAX-DISC1, ZNF391, ZNF74, OSCP1, ZNF501, ZDHHC14, UBTF, ATP6V0A1, TUBG2, SLC25A3, DCBLD2, CPSF4, OCIAD1, NPHP4, C21orf59, NCK2, NUSAP1, TPD52L2, C10orf84, FASN, PLCD3, SEC24A, C1orf101, UBL3, HYAL3, FARSA, ZNF136, ARNTL2, TMEM55B, PNKD, IFFO2, CDK5RAP3, TNNI3K, LRRCC1, GSTZ1, ATPGD1, MINK1, NOP58, DDX58, OCEL1, CMBL, ZNF222, RCBTB2, ASL, MKI67, UGDH, UTP23, MIF4GD, RAP2B, BAT3, HDAC7, FAM164A, SLU7, POLD2, G6PC3, CHMP1B, TTK, CASP6, AAK1, FRY, SUCLG1, PPM1F, TRIM27, CPEB3, CCDC59, DPYD, C16orf72, MTIF2, RUFY2, CCNB2, KBTBD7, SLC35A3, STMN1, TBCB, RBCK1, FTSJD1, C16orf42, RANBP3, NIT2, NAGK, ATRN, FICD, RPUSD3, C15orf61, GPATCH2, NUDT6, RSPH1, EHD4, ALDH9A1, UBE2H, MTR, HDAC11, ELP2, ST8SIA1, ELMO2, TSPAN3, RPAP1, SLMO2, IPO8, GUCD1, TUBB, FANCC, UBOX5, HELQ, PRKCA, USP33, GADD45B, SAMM50, CPEB2, LRPPRC, NSUN6, EIF4A3, PCIF1, ZNF707, SYNPO, HDLBP, DCUN1D4, AHDC1, CCDC93, RASA2, ECHS1, FXR1, ZNF570, FNIP1, GIN1, POLD2, PTCD3, ANKRD11, C14orf21, ELP2P, GSDMD, CDKN2D, CHMP5, SLC10A7, PJA2, C6orf1, TIPIN, DNAJC8, AEN, NEU1, SLC19A2, CDC123, C2orf42, PPP1R3E, CLSPN, PPPDE2, DCBLD1, PLEKHG2, SPAG5, DCP1A, STRAP, ARHGAP12, PSMG1, AMIGO2, ZNF318, MTMR4, ALG8, ZNF259, UBE2J1, DUS4L, MTIF2, CASP8AP2, SPATA6, TAPBP, C14orf80, ZBTB42, INTS5, PCYOX1L, ANAPC13, NUP50, NDUFB5, PLEKHG4, TERF2, RBM15B, APPL1, C19orf40, N6AMT1, KCTD7, UBA52, OSBPL2, PSMB1, PSMD9, MCM4, PDE4C, NME3, RPS6, BRD2, KCNK6, DCAF4, ACOT7, SYNRG, EIF4B, SELO, GPATCH3, BRIX1, TAPBP, RPSA, MEA1, PGAM5, KLHL7-AS1, NT5C3, SRSF7, CLK3, SAMHD1, ELAC2, KIAA0895, WSB1, GANAB, PNPLA8, UNC119B, CCAR1, TTC4, DAAM1, XPO4, MRPL12, NCL, CPPED1, CEP95, SRSF10, PDCD5, C11orf54, NSUN6, SLC35C1, BAT2, MED11, HS1BP3, SETMAR, ANXA4, C12orf10, FAM63A, HERPUD2, LNP1, FAF2, JOSD1, LOC100130987, PEF1, PPP2R2A, ZFP90, ZNF639, ZZEF1, SIX5, MLL5, AATF, VRK1, TM2D1, C20orf11, NARF, PPP1CC, SIDT2, CTNNBIP1, CHCHD2, HCN3, MAP3K6, FAM111A, KCNA3, ERBB2IP, ELL, RPLP2, SEC31A, TNK2, TPRKB, TRAPPC3, TFCP2, PEX7, C7orf30, WRNIP1, IER2, LSG1, GRK4, MOBKL3, TRAM1, KIAA0564, PHF20L1, MDH1B, CEP164, MUL1, PHB2, ZZZ3, BTBD7, SIX5, CAMK2D, NAA38, TRIM45, MRPL55, NEDD4L, C2orf42, MARS2, KIAA1949, MTHFD1, POLE3, EZH2, TSPAN4, SH2D3C, TRIM45, LAPTM4A, SNORA76, ANKRD52, NAGS, ZDHHC5, DENND6A, FTSJD1, DCBLD2, FBXO34, RPP25L, ZBTB7A, IQCH-AS1, LINC00899, MORF4L1, CPEB2, HIST1H2BJ, PPM1B, NDE1, QSER1, FBXO31, LRRC8A, BLCAP, BRI3BP, PRMT1, RPLP1, GNRHR2, METTL23, ZNF490, SRP14, NCAPG, TMEM149, GBAS, MKNK1, H3F3B, ANKRD27, LOC440356, VOPP1, KLHDC2, PODNL1, RAD17, FIS1, SSNA1, SGCE, FKBP3, CHP, ZNHIT6, MIF4GD, FDFT1, GPD1L, ZNF566, MLL, IGF1R, BANP, ZBTB41, MICA, CPSF2, ERO1LB, FOSL1, C19orf61, ABHD13, ATP1A1, SFPQ, C20orf30, ZNF546, YWHAZ, CYB5R3, MFN2, TP53, S100A6, LMAN2, FGFBP3, RFC3, CDC14A, C20orf27, ZNF775, ABHD4, SGSM3, ZNF295, COPB2, PEX26, LOC100506100, DCLRE1A, POLD4, ANKRD11, STX5, LAPTM4A, UNG, NDUFB3, BTG2, HSPA9, RTN4, TLCD1, WDR47, SLC37A4, CPNE8, FZD7, TMEM48, ZNF620, CENPM, C1orf51, STK10, MPHOSPH6, UEVLD, DNAJB12, MTERFD3, TBRG4, NDUFB5, MIR92B, OXNAD1, GPR180, BSCL2, SLC9A8, NDUFV2, RABIF, CD47, DDX46, RBMXL1, CDC7, CDK19, TMEM135, STRAP, ERGIC2, MGC16275, EHD3, ANKRD49, WIZ, DCP1B, KLHL22, PYCR2, PTCD3, ATXN3, ZFYVE19, SMIM14, NDUFA2, FAM55C, SMARCA5, PRDX5, SNHG4, IPO8, KANSL1L, NF2, ATP6V1H, POLD1, RBM15B, PPP6R1, ATG5, PSMB9, LOC400657, AP2M1, XRCC1, INIP, GATAD1, RPL36AL, VPS54, RCHY1, GCH1, ERCC2, MUS81, NDUFA7, SLC35A4, UBXN1, MED7, MYO15B, VPS45, EPB41, PRPF8, LSS, S100A6, BRF1, NAA35, ATP6V0D1, FAM175B, SLC10A7, TIGD1, C19orf61, CNOT8, RAPGEF6, CLN5, DAPK3, STT3A, CDC42EP4, MYH10, XYLB, CDKN2A, MRPL44, BTBD10, EIF4G2, CABIN1, CDR2, FUT8, PPP2R4, HMGN1, CDK1, EVI5, GK5, SNORA76, CPNE1, EDC4, SS18L2, PTBP1, NAA25, TCF12, EME1, GSS, HK1, DFFA, ZNF792, C1orf212, ATP5H, SNX27, KIAA0528, ANXA4, MYADM, GPR113, ATP5G2, NVL, FICD, ZNF582, POLR3H, ABCB6, ZCCHC24, GNS, TMEM18, MIR148A, EPB41, SPOPL, CDNF, ZNF688, KLHDC2, RAP2B, MSRB2, FAM98A, RPL29, ABI1, ZNF784, CDC6, MRPL18, SNRNP70, RAB3GAP2, RAP1A, C9orf163, PINX1, CNST, TRIM41, WEE1, ZNF805, ATAD2B, IMMP2L, WDR24, C6orf173, GOLGA1, MTCH1, PELP1, TMEM44-AS1, STK24, HYAL2, GLUD1, RAB2A, ENO2, POLR2E, TMEM131, RCBTB2, TULP3, MLL5, YOD1, C1orf43, EIF2AK2, CGGBP1, ZNF707, RSU1, GTF3C5, PTEN, TMEM80, ERMAP, CCT2, KIF21A, C6orf204, PSMA5, MXD1, SFT2D2, AACS, USP18, ZNF782, SNORD50B, BAT5, CAB39L, C7orf50, SMC3, E4F1, BAT5, BNIP2, SAPS3, C17orf62, BAG2, BBIP1, EIF6, ERCC4, ERLIN2, GFI1, SLC27A5, LTB, ZNF57, ERO1L, CCDC134, BTG2, LRSAM1, ZCCHC2, C1orf83, GATAD2B, SLTM, RPL15, IP6K1, DENR, ANKRD13A, ZBTB2, SUPT5H, EEF2, ERGIC2, USP21, ZSWIM7, DSCR3, ORC1L, ZNF77, C6orf1, ZFP36L2, TSTD2, SOX4, ETFDH, ANP32B, RAI14, PSMA7, BRUNOL6, ZFP36L2, PGLS, PHF12, MTHFD1L, IFT140, PHTF1, RNASEN, ZNF862, SPC25, SPRY2, ORAI1, ANKRD11, DDX12, TXNRD1, CISD1, EIF3A, NDRG3, CNIH4, MRPS9, TMEM106B, ZNF782, PRDX5, LOC282997, NPHP4, POLE4, LENG8, METTL13, ITPR1, ZNF764, MFAP1, RNFT1, PSMA2, SYNGAP1, TMEM214, KLF10, SCAMP1-AS1, DEPDC1, NAA40, MTA2, PIGT, ARRDC2, BAT5, HSPA13, IL4R, WDR87, ZDHHC14, NSUN3, CDC2, PSMA5, C8orf59, PLEKHF1, ACAD11, PEF1, TCF7, DCTPP1, HDAC1, BAT5, NANP, METAP2, KIAA0652, CLK3, ITCH, NUP160, CCDC150, PPPDE2, BCL2L1, MARS2, CENPK, CHIC2, FZD6, PSMD7, PRKAA1, TMEM194A, C6orf136, MTIF2, ARRB2, RAP1GDS1, SNX33, SCAI, SEC23IP, ARF6, ZZEF1, ATP1A1, PRR3, NBPF3, ZNF169, CBX6, TMEM143, APPBP2, ZNF573, DHDDS, AKAP13, SFRS13A, RPL26, IL12A, KLRAQ1, SERGEF, CCNF, PSMA5, GTF2A1, CPNE8, ZNF578, COX10, LCA5, FAM96B, TMEM14C, ENTPD5, JUNB, PDRG1, MED18, SMIM20, LRIG1, EIF4A2, C10orf4, FABP5, RNASEH2B, LZTFL1, ERMP1, UGDH, MAP1S, HMGN1, RB1, ECI1, ZHX2, CD55, GRSF1, IBTK, LSS, HINFP, PCNA, SH3YL1, STK36, CCDC21, MKL2, PTBP3, NUF2, TMEM68, C4orf19, OSCP1, HNRNPUL1, SYDE2, PDCD5, SAMHD1, NUDT18, PSMD1, MICALL1, NFE2L2, CAPN1, TMEM161A, PSME2, SLBP, ACIN1, MARS, TUBA1A, MTIF3, C22orf25, C20orf7, PDP2, ISM1, EIF1AD, MKL2, ZC3H12A, TMEM251, RAB11B, PPPDE2, MSH5, PDE7A, KLC2, TMEM218, PEX26, PARPBP, CLASP2, MUDENG, ZNF507, DVL1, CD82, YTHDF2, APPL1, AGXT2L2, FNTB, NR4A2, SLC25A28, NFU1, BTBD1, SLC16A13, DNAJB6, ADRB2, HSD17B8, XKR9, CISH, CETN3, THUMPD3, CHST12, CDON, TAPBP, TIMELESS, KIAA2018, NANS, GRPEL2, COQ3, PRR13, NEU3, SRPRB, Mar-08, VCPKMT, LENG1, RNF103, MLF1IP, STARD3, C14orf4, MEF2D, IDH3B, FAM115A, GPX4, HNRNPL, PDP2, PTCD2, LOC93622, IKZF1, ATP2C1, RPL18, TSGA14, RTTN, RPS16, ZNF793, NETO2, SUGT1, ANXA4, NAT14, OGDH, SSRP1, SGSM2, TIPIN, GGCX, KIAA1324, GNS, ACADS, NDUFS8, RNF40, CDK5, MRPL41, LYSMD4, HCFC2, CPEB2, B4GALT3, DNA2, ZNF395, WIPF1, AP1M1, MTURN, MAP4K5, NUP188, ARIH2, STT3B, LOC100272217, TWISTNB DMRs: BAG1, BTG2, CHP1, KIFC1, MKL2, MTR, PEX11B, POLD2, S100A6, SNORD104, SPG7
Moulton et al., 2024 [46]	Two-armed randomised controlled trial	Female first primary BC patients undergoing medical treatment (training group: *n* = 10; control group: *n* = 10) (45–65 years of age)	Blood sample	16 weeks of mixed-modality exercise training 2 times/week	MSP (SOD1, SOD2, Catalase, RASSF1A, L3MBTL1, RASSF1A)	SOD1, SOD2, L3MBTL1
Zeng et al., 2012 [10]	Randomised clinical trial	Physically inactive, postmenopausal female BC patients (*n* = 12) (56.5 ± 9.5 years of age) Breast tumour samples (*n* = 348) (111 ≤ 50, 139 = 50–65, 98 = ~65 years of age)	Blood sample Breast tumour sample	6 months of moderate-intensity exercise (150 min/week)	Infinium HumanMethylation27 BeadChip: Panel of 14 495 genes MSP: (L3MBTL1)	EPS15, DYDC1, WNT7A, SULF1, KPNA5, AQP5, ALG1, C1R, PARP11, INSRR, CDC26, ZNF222, PPP2R3A, TMEM100, IFT172, C8orf53, CXCL10, NALP11, HINT2, OSTF1, ERVK6, DC-UbP, RASA1, DCC RP11-450P7.3, KIAA0980, RBM10, PLAGL1, MEG3, ORM2, DYNC1I1, GAB1, ABCB1, SLC9A7, LRRC14, L3MBTL1, MSX1, PCTK3, BCL2L11, WNK3, GLUD1, MGC39633, PLCZ1 L3MBTL1
**Self-reported PA studies**						
**Author**	**Study design**	**Population**	**Sample**	**Exercise/PA behaviour measured**	**Methylation detection technique and analysed target**	**Significant modulation(s) in DNA methylation as a Result of PA/exercise**
McCullough et al., 2015 (specific) [44]	Population-based-case–control study	Female first primary BC patients (*n* = 532) (20–98 years of age, mean age 59.6)	Breast tumour sample	RPA	-MSP: ESR1, PR, BRCA1 -MethyLight assay: APC, CDH1, CCND2, DAPK, GSTP1, HIN, P16, RARB, RASSF1A, TWIST1	GSTP1
McCullough et al., 2015 (global) [49]	Population-based-case–control study	Postmenopausal female first primary BC (*n* = 1300) (20–98 years of age)	Blood sample	Postmenopausal RPA	LUMA global DNA methylation assay: LUMA Pyrosequencing-based methylation assay: LINE-1	Global DNA methylation
McCullough et al., 2017 [47]	Population-based-case–control study	Female first primary BC patients (*n* = 807) (20–98 years of age)	Breast tumour sample Blood sample	RPA	Gene-specific: -MSP: ESR1, PR, BRCA1 -MethyLight assay: APC, CDH1, CCND2, DAPK, GSTP1, HIN, P16, RARB, RASSF1A, TWIST1 Global: -LUMA global DNA methylation assay: LUMA -Pyrosequencing-based methylation assay: LINE-1	APC, CCND2, HIN, TWIST1

BC, breast cancer; PA, physical activity; MSP, methylation-specific PCR; DMR, differentially methylated region; DMP, differentially methylated position.

## Data Availability

All data obtained from the bioinformatic analysis can be found in the Appendix A.

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
