# Peer review of "Impact of Physical Activity on DNA Methylation Signatures in Breast Cancer Patients: A Systematic Review with Bioinformatic Analysis"

_cancers, 2024, doi:10.3390/cancers16173067_

Round 1

Reviewer 1 Report

Comments and Suggestions for Authors

Review: Impact of physical activity on DNA methylation signatures in breast cancer patients: A systematic review with bioinformatic analysis

In their manuscript, the authors review literature on the impact of physical activity (PA) on DNA methylation in breast cancer (BC) patients. While a huge number of publications touch this topic, the authors identified only six that present original data. Notably, they excluded all publications, ‘where PA levels/exercise were not identified as independent modulators of DNA methylation.’ Here, I hope that this does not mean that they excluded all negative results. In this case, I would recommend an immediate rejection due to biased research.  
Let us for the moment assume that this means that these articles provide ‘no clear analysis between PA and methylation’ (see Fig. 1). Even in this case, I have several serious concerns.
INTRODUCTION
The authors note that ‘PA significantly impacts DNA methylation patterns’, but do not provide numbers. This gives the misleading impression that it would reverse BC specific methylation profiles. Actual, about 1% of the differentially methylated position (DMPs) in BC are affected (Gorski et al.), meaning a tiny fraction.  Moreover, the authors state that they would ‘comprehensively’ study the effects of PA-induced DNA methylation on different biological aspects in BC patients using bioinformatic analysis. Actually, they do some basic enrichment analysis, based on questionable assumptions (see below).
METHODS AND MATERIALS
See my first comment.
RESULTS
As a summary of the literature search, the authors provide a table (Tab.1) that characterizes the 6 selected studies. In the last column (Significant modulation(s) in DNA methylation as a result of PA/exercise) they provide gene names instead of DMPs. This is also misleading. These genes are just associated with the DMPs, but only a tiny fraction shows differential expression. A helpful annotation would inform about the DMPs and their position (promoter, body etc.).  Such annotation is given e.g. by Gorski et al. in very much detail but is missing in other studies e.g. Zeng et al. Thus, I suggest to provide just the name of those genes that change expression under PA induced aberrant methylation.  This information is missing.
Further missing information:
The studies all analyze differential methylation. Do they use the same definition, e.g. about the minimum methylation difference?
Three studies analyze blood samples. Did they correct their data for blood cell composition?  E.g. L3BMTL1 shows lineage specific methylation. Although the differences are not large (up to 10% T-cell vs. myeloid), this opens the question whether PA changes the blood composition only.
Regarding gene-specific promoter methylation studies, what is the definition of the promoter?
In my opinion, this information is required in order to verify the results. Which studies provide their data, e.g. for comparative purposes?

BIOINFORMATICS PATHWAY ANALYSIS
In this section, the authors do some enrichment analysis for the genes that are associated with the DMPs found in the studies.  What is the rational? Only a tiny part of these genes changes expression in correlation with the methylation changes. In worst case, not one of the highlighted pathways becomes activated or repressed following PA.  Gorski et al. already provided similar analysis (see text and supplement 1C). What is the benefit of these parallel studies?

DISCUSSION
The discussion bases largely on, and overstresses the critical pathway analysis and thus represent in large parts speculation. No question that PA exerts beneficial effects within BC patients. My impression after reading this review is that the link to epigenetics is still weak. Thus, I suggest focusing on the potential ‘Mechanisms Underlying PA-Induced DNA Methylation’. Best with an illustrative figure. All the nice future work will be more efficient if a clear hypothesis exist.  
Unforgivable: ‘Our findings indicate that PA can influence both global and gene-specific DNA methylation, potentially contributing to improved clinical outcomes for BC patients.’

In summary, the manuscript requires a major revision.

Author Response

We have really appreciated the suggestions from the Reviewer because he/she helped us to possibly improve the quality of our manuscript. We went through all suggestions offered by Reviewer as detailed in the point-by-point response to his/her comments

Point-to-point response

Reviewer 1

Comments and Suggestions for Authors

Review: Impact of physical activity on DNA methylation signatures in breast cancer patients: A systematic review with bioinformatic analysis

Q1) In their manuscript, the authors review literature on the impact of physical activity (PA) on DNA methylation in breast cancer (BC) patients. While a huge number of publications touch this topic, the authors identified only six that present original data. Notably, they excluded all publications, ‘where PA levels/exercise were not identified as independent modulators of DNA methylation.’ Here, I hope that this does not mean that they excluded all negative results. In this case, I would recommend an immediate rejection due to biased research.  

A1) We thank the Reviewer for this comment because it allows us to better explain our selection criteria of articles published in this field. As we stated in the section 2.1 (Study selection), for the choice of articles, we relied on very rigorous selection criteria that allow us to evaluate in the best way the impact of physical activity on methylation levels in BC patients. To date, many works have been published in this field, many of these are in healthy subjects where the exercise intervention is not the only component examined. The selection of these articles would not allow to clearly identify what could be the contribution of physical activity compared to that induced by other components (e.g., food supplements, specific dietary plans, etc.). This led to the identification of the 6 articles that are indicated in the manuscript, without excluding any other work. However, we agree with the reviewer that a review of the literature based on the results (e.g., only the positive ones) is not ethically correct and all works that operate in this way should be rejected.

To avoid misleading concept, we have reworded the sentence accordingly.

Q2) Let us for the moment assume that this means that these articles provide ‘no clear analysis between PA and methylation’ (see Fig. 1). Even in this case, I have several serious concerns.
INTRODUCTION
The authors note that ‘PA significantly impacts DNA methylation patterns’, but do not provide numbers. This gives the misleading impression that it would reverse BC specific methylation profiles. Actual, about 1% of the differentially methylated position (DMPs) in BC are affected (Gorski et al.), meaning a tiny fraction. 

A2) We thank the reviewer for this comment. In order to be more precise, we have provided the numbers, as much as is available and where relevant, alongside the methylation results to provide more context to the findings.

Q3) Moreover, the authors state that they would ‘comprehensively’ study the effects of PA-induced DNA methylation on different biological aspects in BC patients using bioinformatic analysis. Actually, they do some basic enrichment analysis, based on questionable assumptions (see below).

A3) We would like to thank the reviewer for pointing this out. We have changed the phrasing of the statement to represent our analysis more accurately. We address the rest of the points regarding bioinformatic analysis below (A9).

Q4) METHODS AND MATERIALS
See my first comment.

A4) Addressed above at the first comment.

Q5) RESULTS
As a summary of the literature search, the authors provide a table (Tab.1) that characterizes the 6 selected studies. In the last column (Significant modulation(s) in DNA methylation as a result of PA/exercise) they provide gene names instead of DMPs. This is also misleading. These genes are just associated with the DMPs, but only a tiny fraction shows differential expression. A helpful annotation would inform about the DMPs and their position (promoter, body etc.).  Such annotation is given e.g. by Gorski et al. in very much detail but is missing in other studies e.g. Zeng et al. Thus, I suggest to provide just the name of those genes that change expression under PA induced aberrant methylation.  This information is missing.

A5) Unfortunately, as not every study analyses gene expression, it would not be possible to include this information for all of our included studies. Also, as the focus of this systematic review was primarily to look at the effects on DNA methylation, we have provided this information in terms of genes (majorly considering the methylation trends within promoter regions) as named by the authors of the studies to provide a cohesive set of results. As mentioned by the reviewer, DMPs and positions are not available in all studies, and as gene expression has also not been analysed in all studies, we have presented the results as the gene associated with the methylation change as indicated by the authors.

We have included this lack of information in the limitations section, as well as directions for future research to encourage this information should be prioritised in future publications.

Further missing information:

Q6) The studies all analyze differential methylation. Do they use the same definition, e.g. about the minimum methylation difference?

A6) We thank the reviewer for this question. Unfortunately, most included studies do not include or state their definitions regarding cutoffs or minimums for the methylation changes. Some information is given across studies; however, it is not given by all. For example, McCullough et al. (2017) specified a 4% cutoff to dichotomize into methylated vs unmethylated, and the necessary calculation to compute methylation levels is given by most dependent on the analysis method. In addition, considering the variety in methylation analysis methods used, each which may have differences in sensitivity, if such a definition was given, they may have varied between each other in any case to fit the methods used in each study. Most studies interpreted the changes in methylation based on statistical significance and the necessary considerations relevant to their chosen analysis methods.

To make things clearer and highlight this aspect in our manuscript, we included this information and related references in the section 3.2. Studies Characteristics.

Q7) Three studies analyze blood samples. Did they correct their data for blood cell composition?  E.g. L3BMTL1 shows lineage specific methylation. Although the differences are not large (up to 10% T-cell vs. myeloid), this opens the question whether PA changes the blood composition only.

A7) We thank the reviewer for this comment. It is known that the magnitude of the exercise-induced changes in hematological parameters depends on current external conditions (temperature, humidity), the type, intensity and duration of exercise, as well as the level of physical fitness of the examined person. All articles included in this systematic review do not report any information about the correction of blood cell composition data. Indeed, to date, many studies conducted on sedentary populations report contrasting values (DOI: 10.7205/milmed.170.7.590; DOI:10.11114/jets.v6i8.3374; DOI: 10.1136/bjsm.2005.022095; DOI: 10.1152/japplphysiol.00875.2004). However, similarly to many of the published works mentioned above, and given the type of training proposed in these studies and the medium-low intensity of the training sessions, we believe that the variation of the blood cell content as well as the plasma volume variation is not significant between the beginning and the end of the training period. Moreover, most of the authors verified their result even with qMSP and with the related changes in gene expression in tumor samples.

To avoid possible lack of information, we have added a sentence in the “Study limitations” and “Future Implications for research” sections, to encourage this type of correction in future.

Q8) Regarding gene-specific promoter methylation studies, what is the definition of the promoter? In my opinion, this information is required in order to verify the results. Which studies provide their data, e.g. for comparative purposes?

A8) In gene-specific promoter methylation studies, the promoter is typically defined as the region of DNA located just upstream (5' direction) of a gene's transcription start site (TSS). This region is crucial because it contains binding sites for transcription factors and other regulatory elements that control the initiation of transcription. Promoter regions often contain CpG islands, which are stretches of DNA rich in cytosine and guanine nucleotides; methylation of these CpG sites can suppress gene expression by preventing the binding of transcription factors.

So, in the context of methylation studies, the promoter refers to this regulatory region where DNA methylation can influence whether the associated gene is actively transcribed or silenced.

To clarify this aspect, we have inserted a sentence in the section “2.2 Data collection process”. However, as usually happens for many systematic reviews, all the data we used and presented in our manuscript are those present in published works and easily accessible by all researchers.

BIOINFORMATICS PATHWAY ANALYSIS

Q9) In this section, the authors do some enrichment analysis for the genes that are associated with the DMPs found in the studies.  What is the rational? Only a tiny part of these genes changes expression in correlation with the methylation changes. In worst case, not one of the highlighted pathways becomes activated or repressed following PA.  Gorski et al. already provided similar analysis (see text and supplement 1C). What is the benefit of these parallel studies?

A9) We thank the reviewer for this question. By conducting this enrichment analysis, it allows us to explore the broader functional implications of DNA methylation changes, even if not all of these changes have directly been shown to translate to altered gene expression. By analyzing the full spectrum of methylation patterns across all relevant studies, we can identify potential pathways that might be modulated by physical activity, contributing to the overall biological response in BC recovery. Furthermore, while Gorski et al. performed a similar analysis, our enrichment analysis is distinct in that it is done on the collected data from all the studies included in our review. The goal is to show the overall picture in what happens due to PA-induced DNA methylation in BC survivors as the literature, based on the articles which meet our selection criteria (PA and DNA methylation in BC patients), has shown up to the present moment. This provides a more comprehensive and cohesive understanding of the functional changes potentially driven by PA, available in the literature, offering insights that may not have been captured by individual studies. Therefore, our analysis adds value by synthesizing and contextualizing the existing data to present a more complete picture of the potential epigenetic mechanisms at play.

DISCUSSION

Q10) The discussion bases largely on, and overstresses the critical pathway analysis and thus represent in large parts speculation. No question that PA exerts beneficial effects within BC patients. My impression after reading this review is that the link to epigenetics is still weak. Thus, I suggest focusing on the potential ‘Mechanisms Underlying PA-Induced DNA Methylation’. Best with an illustrative figure. All the nice future work will be more efficient if a clear hypothesis exist.  

A10) We would like to thank the reviewer for this suggestion. We have added a new figure (Figure 3) in the discussion to better illustrate the potential mechanisms underlying PA-induced DNA methylation.

Q11) Unforgivable: ‘Our findings indicate that PA can influence both global and gene-specific DNA methylation, potentially contributing to improved clinical outcomes for BC patients.’

A11) We would like to thank the reviewer for this comment and apologise. We have reworded the phrase, specifically by changing the ‘Our findings indicate..’ to not take ownership of the findings from the authors of the studies we included in this review.

Reviewer 2 Report

Comments and Suggestions for Authors

This systematic review summarizes the available evidence on the impact of physical activity on DNA methylation in breast cancer patients. The topic is analyzed both at the level of global DNA methylation and gene-specific DNA methylation. The general conclusion of the work suggests a possible positive effects of physical activity on clinical outcomes for BC patients.

The work is written clearly and comprehensibly, with appropriately cited literature sources, and it meets the characteristics of a systematic review. One of the strengths of the work is the bioinformatic analysis of the results from the processed literature sources. However, I would point out the small number of sources processed, which results from the selected schema of inclusion criteria. Only 6 publications were included in the review, which were further divided into smaller subsets (global DNA methylation, gene-specific DNA methylation). At the level of global DNA methylation, only two publications were analyzed, and their results were not entirely consistent with each other. In the case of gene-specific DNA methylation, 5 studies were analyzed. The work lacks a slightly more critical perspective on the analyzed studies.

My questions/comments to the authors:

1) The selection of articles followed a generally correct and logical procedure, but it resulted in a relatively small number of articles meeting the required criteria. For example, for the analysis of global methylation, only two sources could be used, leading to some mutually incoherent observations. Have the authors considered modifying the selection mechanism to potentially increase the number of suitable sources—for instance, for the analysis of global vs. gene-specific DNA methylation?

2) Could the authors clarify the reason for excluding 145 sources in the second step of the selection process? In the text on line 165, it is stated that 26 articles were initially selected based on the title and abstract. This formulation suggests that these 26 sources were chosen as part of some preliminary analysis, but no additional articles were added to the analysis afterward. If only these 26 sources met the required criteria, I would recommend rephrasing this sentence.

3) One of the factors that may significantly influence the level of DNA methylation, as well as the course of the disease and treatment, is the pre-disease fitness level. As the authors mentioned, Gorski et al. observed significant changes in DNA methylation in skeletal muscle between healthy controls and BC patients. It would certainly be appropriate to analyze differences within the group of BC patients themselves concerning their pre-diagnostic training status. If these exercise-induced DNA methylation modifications were stable for up to 10 years, it is relevant to expect differences in DNA methylation status between physically active and inactive BC patients during and after treatment.

4) Can the authors provide more details on the demographic background of BC patients included in the analyzed studies? Age is likely another significant factor when comparing changes in DNA methylation. For example, McCullough et al. includes postmenopausal BC patients, but as is unfortunately known, breast cancer also affects much younger age groups.

5) Did the authors come across any potential negative effects of physical activity on the health status or treatment course of BC patients during their literature review? Despite the many documented benefits associated with physical activity in oncology patients, some physicians do not recommend or directly advise patients to avoid physical activity due to the potential strain on an already weakened body.

Summary:

The publication addresses a current issue, and the findings are processed in a professionally correct and comprehensive manner. I also consider the bioinformatic analysis as a significant contribution. I would recommend the authors consider further analyzing some of the discrepancies between the results of the reviewed studies. However, in general, I recommend the publication of this systematic review.

Author Response

We thank the Reviewer for her/his general positive comments. We have really appreciated the suggestions from the Reviewer because he/she helped us to possibly improve the quality of our manuscript. We went through all suggestions offered by Reviewer as detailed in the point-by-point response to his/her comments

Point-to-point response

Reviewer 2

The work is written clearly and comprehensibly, with appropriately cited literature sources, and it meets the characteristics of a systematic review. One of the strengths of the work is the bioinformatic analysis of the results from the processed literature sources. However, I would point out the small number of sources processed, which results from the selected schema of inclusion criteria. Only 6 publications were included in the review, which were further divided into smaller subsets (global DNA methylation, gene-specific DNA methylation). At the level of global DNA methylation, only two publications were analyzed, and their results were not entirely consistent with each other. In the case of gene-specific DNA methylation, 5 studies were analyzed. The work lacks a slightly more critical perspective on the analyzed studies.

My questions/comments to the authors: 

Q1) The selection of articles followed a generally correct and logical procedure, but it resulted in a relatively small number of articles meeting the required criteria. For example, for the analysis of global methylation, only two sources could be used, leading to some mutually incoherent observations. Have the authors considered modifying the selection mechanism to potentially increase the number of suitable sources—for instance, for the analysis of global vs. gene-specific DNA methylation?

A1) We thank the reviewer for this comment because it allows us to better explain the focus of this systematic review.

To date, the field of exercise oncology is extremely heterogeneous from various points of view such as the characteristics of the subjects, the subtype of tumor, tissue analysed, the period/type of physical activity (PA), whether it is done in primary, secondary or tertiary prevention, biological parameters analysed, as well as concomitance with other interventions (e.g., nutritional, food supplements, etc.), Therefore, if on the one hand the stratification of the above mentioned variables leads to a still very limited number of published studies, on the other hand putting together many of these studies without a specific stratification would give unclear information.

Our research group is strongly interested in analysing the impact of physical activity in subjects affected by breast cancer, especially during the period of medical treatment. In fact, to date we have already published several articles in this time window (doi: 10.1016/j.redox.2024.103033; doi: 10.3390/antiox12051138; doi:10.3390/ijms25168596), where we have identified potential benefits induced by structured physical activity compared to those who do not perform any structured physical activity.

As we stated in our systematic review, we want to highlight the impact of PA and DNA methylation in BC patients, giving an overview of what has been done in the research field.

We tried to consider the possibility of modifying the selection mechanism to potentially increase the number of suitable sources, but most of the published works have been done on healthy populations not affected by breast cancer (primary and/or secondary prevention). However, given the increasing number of breast cancer cases, it seems right to present, with all the possible limitations, what has been done to date in the world of research on this specific population (tertiary prevention).

Q2) Could the authors clarify the reason for excluding 145 sources in the second step of the selection process? In the text on line 165, it is stated that 26 articles were initially selected based on the title and abstract. This formulation suggests that these 26 sources were chosen as part of some preliminary analysis, but no additional articles were added to the analysis afterward. If only these 26 sources met the required criteria, I would recommend rephrasing this sentence.

A2) We thank the Reviewer for this comment and in agreement with her/him we create a new paragraph (3.1 Search Results) where we better explain the selectin process.

Q3) One of the factors that may significantly influence the level of DNA methylation, as well as the course of the disease and treatment, is the pre-disease fitness level. As the authors mentioned, Gorski et al. observed significant changes in DNA methylation in skeletal muscle between healthy controls and BC patients. It would certainly be appropriate to analyze differences within the group of BC patients themselves concerning their pre-diagnostic training status. If these exercise-induced DNA methylation modifications were stable for up to 10 years, it is relevant to expect differences in DNA methylation status between physically active and inactive BC patients during and after treatment.

A3) We agree with the reviewer and for this reason we have included in section "3.2 Characteristics of the studies" the information regarding the baseline physical activity level for each selected study, comparing the possible differences/similarities between them.

Q4) Can the authors provide more details on the demographic background of BC patients included in the analyzed studies? Age is likely another significant factor when comparing changes in DNA methylation. For example, McCullough et al. includes postmenopausal BC patients, but as is unfortunately known, breast cancer also affects much younger age groups.

A4) We thank the reviewer for highlighting this very important aspect. In agreement with the Reviewer, we took into account the age of the subjects recruited in each study and compared the possible differences/equalities (see section “3.2. Studies Characteristics”).

Q5) Did the authors come across any potential negative effects of physical activity on the health status or treatment course of BC patients during their literature review? Despite the many documented benefits associated with physical activity in oncology patients, some physicians do not recommend or directly advise patients to avoid physical activity due to the potential strain on an already weakened body.

 A5) To our knowledge, and on the basis of our research experience in this field, all studies that include an exercise intervention design a specific and tailored training protocol. Given the significant impact that physical activity has on the course of the disease, in many countries, including Italy, physical activity is being introduced into the therapeutic pathways of several non-communicable diseases. Furthermore, all patients are medically screened to verify their suitability for the training program. To date, given the way the field of motor sciences adapted to the physio-pathological state of the individual has evolved, it is very difficult to detect its negative effects. During our review of the literature, we did not encounter articles highlighting a negative effect of physical activity on this type of patients. All studies selected did not report patient dropout due to adverse effects induced by physical activity and/or negative outcomes.
